# Morphological Characters and Molecular Phylogeny Reveal Three New Species of Subgenus *Russula* from China

**DOI:** 10.3390/life12040480

**Published:** 2022-03-25

**Authors:** Bin Chen, Junfeng Liang, Xumeng Jiang, Jie Song

**Affiliations:** 1Institute of Biological and Medical Engineering, Guangdong Academy of Sciences, Guangzhou 510316, China; binsanity12@caf.ac.cn; 2Research Institute of Tropical Forestry, Chinese Academy of Forestry, Guangzhou 510520, China; 3College of Forestry, Northwest A&F University, Yangling 712100, China; jiang_xm1029@nwafu.edu.cn; 4Department of Horticulture and Food, Guangdong Eco-Engineering Polytechnic, Guangzhou 510520, China; jsong@caf.ac.cn

**Keywords:** Russulaceae, phylogeny, taxonomy, 3 new taxa

## Abstract

Three new species are described and illustrated here based on morphological evidence and phylogenetic analysis from China. *Russula*
*leucomarginata* is recognized by a yellowish red to reddish brown pileus center, a yellowish white to reddish white and sometimes cracked margin, and a reddish white to pastel pink stipe. *Russula* *roseola* is characterized by its reddish white to ruby red pileus center, pink to rose margin, adnate to slightly decurrent lamellae with unequal-length lamellulae, reddish white to pink stipe, and occasionally three-celled pileocystidia. *Russula subsanguinaria* is morphologically characterized by a reddish brown to dark brown pileus center, a reddish orange to brownish red margin with striation, a reddish white to pink stipe with an expanded base, basidiospores with moderately distant to dense amyloid warts, and hymenial cystidia turning to reddish black in SV. In this study, we performed phylogenetic analysis based on ITS sequence and 28S-*RPB1*-*RPB2*-mtSSU datasets. Detailed morphological features and phylogenetic analysis indicate that these three new species belong to *Russula* subg. *Russula*.

## 1. Introduction

*Russula* Pers. is a cosmopolitan group and occurs across a wide range of habitats, from the arctic tundra to tropical forests, and forms ectomycorrhizal relationships with diverse host plants [1,2]. It is the largest of all genera in Russulaceae, which includes eight subgenera and at least 2000 species within the genus [3,4,5,6]. Some species of *Russula* are important edible fungi and are also commercially traded around the world [2,7,8,9]. According to the recent statistics on resource diversity of Chinese macrofungi, there are 78 edible species and 30 medicinal species in China [10]. The precise identification of *Russula* is all-important for the development and utilization of these fungi.

Due to the high diversity of morphological features of *Russula* basidiomata, establishing an accurate taxonomy system is challenging. Many systems proposing an infrageneric classification of the genus *Russula* have been proposed in the past few years [11,12,13]. Recent molecular studies based on a representative sampling of the world’s diversity have revealed eight subgenera within the genus: *R.* subg. *Glutinosae* Buyck and X.H. Wang, *R.* subg. *Archaeae* Buyck and V. Hofst., *R.* subg. *Compactae* (Fr.) Bon, *R.* subg. *Crassotunicatae* Buyck and V. Hofst., *R.* subg. *Heterophyllidiae* Romagnesi, *R.* subg. *Malodorae* Buyck and V. Hofst., *R.* subg. *Brevipedum* Buyck and V. Hofst., and *R.* subg. *Russula* [3,4]. Subgenus *Russula* was further divided into two parts in Buyck et al. [3]. The new infrageneric classification system of *Russula* based on a multi-locus phylogenetic analysis was followed in this study.

Recently, some new species of subgenus *Russula* have been successively reported in Asia [14,15], indicating that many unknown *Russula* species are waiting to be discovered. To better understand the biological diversity and geographic distribution of Chinese subgenus *Russula*, extensive surveys have been undertaken in different parts of China. During the investigation and sampling, several interesting specimens were found. Morphological characters and phylogenetic analysis consistently supported their independence from other known *Russula* species. Here, three taxa are proposed as new to science, based on detailed morphological descriptions coupled with illustrations and phylogenetic relations.

## 2. Materials and Methods

### 2.1. Morphological Study

The macromorphological characteristics of collections were photographed and annotated in the field. Specimens were dried at 45–55 °C and deposited in the herbarium of the Research Institute of Tropical Forestry, Chinese Academy of Forestry (RITF). Macromorphological characters were based on detailed notes and photographs of fresh basidiomata. Descriptive terms follow widely used mycological glossaries [16]. Color codes and terms are mostly from Kornerup and Wanscher [17]. Micromorphological features were described using the description templates of Adamčík et al. [5]. All were performed from dried specimens with a ZEISS Imager M2. Basidiospores were observed and measured in Melzer’s reagent in side view excluding ornamentation. Other microscopic structures were observed and measured in Congo red after pretreatment in 5% potassium hydroxide (KOH). Estimates of spore ornamentation density followed Adamčík and Marhold [18]. The hymenial cystidia density estimates refer to Buyck [19]. Pileipellis was examined in Cresyl blue to verify the presence of ortho- or metachromatic reactions [20]. Sulfovanillin (SV) was used to observe the color changes of cystidia contents [21]. The ornamentation and structure of basidiospores were illustrated under a scanning electron microscope (SEM-JEOL JSM-6510). Basidiospore measurements are indicated as (Min–)AV-SD–AV–AV+SD(–Max), where Min = the minimum value, Max = the maximum value, AV = average, SD = standard deviation, and Q stands for length/width ratio of the basidiospores.

### 2.2. Molecular Study

Genomic DNA was extracted by an improved CTAB protocol from dry specimens [22]. In the present study, five loci were amplified and sequenced: 600 base pairs of the ITS region of rDNA using primers ITS1 and ITS4 [23]; 900 base pairs of the 28S nuc rDNA (28S) with primers LR0R and LR5 [24]; 1300 base pairs of the largest subunit of the RNA polymerase II (*RPB1*) using primers RPB1-AF [25] and RPB1-CR [26]; 700 base pairs of the second largest subunit of the RNA polymerase II (*RPB2*) with primers bRPB2-6f and fRPB2-7cr [26,27]; and 600 base pairs of the ribosomal mitochondrial small subunit (mtSSU) using primers MS1 and MS2 [23]. The amplified PCR products were subsequently sequenced on an ABI 3730 DNA analyzer using an ABI BigDye 3.1 terminator cycle sequencing kit (Shanghai Sangon Biological Engineering Technology and Services CO., Ltd., Shanghai, China). The newly generated sequences were submitted to GenBank (Table 1).

### 2.3. Phylogenetic Analysis

Species in the subg. *Russula* core clade with high similarity to our new species and representative species were selected for phylogenetic analyses. Accession numbers for ITS sequences are shown in Figure 1. Accession numbers and references of the sequences used in the multigene phylogenetic tree are given in Table 1. Initial sequence alignment was performed by MAFFT 7.0 (http://mafft.cbrc.jp/alignment/server/ (accessed on 1 March 2022)). To obtain reliable and reasonable results, the online program Gblocks 0.91b (Gblocks_server.html) was used in default parameters; approximately 77.8% of sites retained after Gblock analysis. The final sequence alignment was deposited in TreeBASE (http://purl.org/phylo/treebase/phylows/study/TB2:S28485 (accessed on 1 March 2022)). Maximum Likelihood (ML) and Bayesian Analysis (BA) were implemented for phylogenetic analyses. ML analysis was performed using RAxML-HPC2 on XSEDE (8.2.12) through the Cipres Science Gateway (www.phylo.org (accessed on 1 March 2022)). The analysis was executed by applying the rapid bootstrap algorithm with 1000 replicates to affirm the consistency of the results under the GAMMA model. Only the maximum-likelihood best tree from all searches was kept. For BA analyses, the GTR model was selected as the best substitution model by MrModeltest [29]. BA was carried out with MrBayes on XSEDE (3.2.7a) through the Cipres Science Gateway (www.phylo.org (accessed on 1 March 2022)) under the GTR model. Four Markov chains were run for a total of 50,000,000 generations and trees were sampled every 100 generations, the first 25% of each sampled topology being discarded as part of the burn-in procedure. Bootstrap Support (BS) ≥ 70% was considered significantly supported. Bayesian Posterior Probability (PP) values were calculated from the 50% majority rule consensus trees, and PP ≥ 95% were regarded as significant.

## 3. Results

### Phylogeny

The ITS and 28S-RPB1-RPB2-mtSSU datasets were used in Maximum Likelihood and Bayesian Analysis. Both the ML and BA analyses resulted in essentially the same tree topologies, and only the ML tree is shown in Figure 1 and Figure 2. BA posterior probabilities are also shown along the branches.

The ITS phylogenetic analysis showed that subg. *Russula* obtained a high support (BS 100%, PP 1). The samples of the three new species *R. leucomarginata*, *Russula roseola*, and *R. subsanguinaria* formed strongly supported clades (BS 100%, PP 1.00) and were clearly distinct from known species of subg. *Russula*. *Russula leucomarginata* clusters together with *R. rhodocephala* Bazzicalupo, D. Miller and Buyck. by 83% bootstrap support and forms a sister clade to *R. subsanguinaria* without support. *Russula sanguinaria* (Schumach.) Rauschert is sister to a clade comprising *R. leucomarginata*, *R. rhodocephala*, and *R. subsanguinaria* with 1.00 posterior probabilities. *Russula roseola* clusters together with *R. americana* (Singer) Singer, *R. thindii* K. Das and S.L. Mill., *R. salishensis* Bazzicalupo, D. Miller and Buyck., *R. queletii* Fr. and *R. fuscorubroides* Bon with significant support from MLBS (=79%) and PP (=1).

For multigene phylogenetic analysis, the results were similar to those of the ITS phylogenetic analysis. Subg. *Russula* formed a well-supported monophyletic group. The three new species were clearly separated from the known species.

## 4. Taxonomy

***Russula leucomarginata*** B. Chen, J. F. Liang and X. M. Jiang, sp. nov. (Figure 3A,B, Figure 4A,B, Figure 5 and Figure 6)

MycoBank: MB838150

Diagnosis: Recognized by its cracked pileus, yellowish red to reddish brown pileus center, yellowish white to reddish margin, and reddish white to pastel pink stipe. It is mainly separated from similar-looking species by the internal transcribed spacers (ITS) sequence data. The similarity is less than 97% with all these species.

Etymology: Leuco (Latin) = white; marginata (Latin) = margin; named after the yellowish white to reddish white margin of pileus.

Type: China, Yunnan province, Shangri-La City, Xiaozhongdian town, Shangjisha, in mixed forests of *Abies* and *Picea*, 3282 m.a.s.l., 27°27′32.01″ N, 99°48′52.44″ E, 22 August 2014, leg. Zhao2204 (RITF3133, holotype), GenBank number ITS: MW301626.

Description: Basidiomata medium-sized. Pileus 50–80 mm in diam., first planoconvex, becoming applanate with a slightly depressed center after maturation; margin slightly striated, sometimes cracked; cuticle dry, smooth, glabrous, peeling readily; yellowish red (8B8) to reddish brown (9E8) in the pileus center; margin yellowish white (4A2) to reddish white (9A2) with tawny tinge in some parts. Lamellae adnate, 2–6 mm in height, moderately distant, brittle, white (1A1) to yellowish white (2A2); lamellulae often present and irregular in length, furcations absent or rare, edge entire and concolorous. Stipe 40–70 × 12–22 mm, subcylindrical or obclavate, central, subcylindrical to cylindrical, slightly expanded towards the base, rugulose longitudinally, reddish white (8A2) to pastel pink (9A4), medulla hollow. Context 2–4 mm halfway to the margin in pileus, white, unchanging; taste slightly acrid; odor indistinct. Spore print yellowish.

Basidiospores (6.2–)7.0–7.7–8.4(–9.3) × (5.5–)6–6.5–7(–7.7) μm, Q = (1.03–)1.09–1.18–1.27(–1.38), subglobose to broadly ellipsoid; ornamentation of large, moderately distant to dense (6–8(–9) in a 3 μm diam. circle) amyloid warts, 0.5–0.8(–1) μm high, mainly isolated, occasionally fused in short chains ((0–)1–2) in the circle), rarely connected by line connections (0–1(–2) in the circle); suprahilar spot large, amyloid. Basidia (31–)34.8–38.3–42.2(–47.3) × (9.3–)10.7–12.2–13.6(–15.5) μm, mostly 4-spored, some 2- and 3-spored, clavate; basidiola clavate or ellipsoid, ca. 6–15 μm wide. Hymenial cystidia on lamellae sides moderately numerous, ca. 950–1200/mm^2^, (46.3–)58.3–79.2–100(–123) × (8.6–)10.2–12.4–14.6(–18.5) μm, fusiform or cylindrical, apically often obtuse, occasionally acute, sometimes with 2–6 μm long appendage, thin-walled; contents abundant granulose, turning to reddish black in SV. Hymenial cystidia on lamellae edges often smaller, (42.4–)50.8–61.2–71.6(–85.7) × (6.3–)8.5–9.8–11.2(–13.2) μm, cylindrical, clavate or lanceolate, apically usually obtuse, occasionally acute, sometimes with 3–8 μm long appendage, thin-walled; contents granulose or crystalline, turning to reddish black in SV. Marginal cells (11–)14.8–21.2–27.5(–32.8) × (3.9–)4.2–5.5–6.9(–9.4) μm, cylindrical or clavate, often flexuous. Pileipellis orthochromatic in Cresyl blue, not sharply delimited from the underlying context, 120–150 μm deep, two-layered; suprapellis 60–80 μm deep, composed of ascending, repent, loose hyphal terminations; subpellis 70–80 μm deep, composed of strongly gelatinized horizontally oriented narrow hyphae. Hyphal terminations near the pileus margin rarely branched, often flexuous, thin-walled; terminal cells (16.8–)20–27.7–35.4(–45.2) × (3.2–)3.6–4.7–5.9(–7.5) μm, mainly cylindrical, occasionally clavate, apically usually obtuse, or occasionally constricted; subterminal cells often wider, usually unbranched, ca. 3–7 μm wide. Hyphal terminations near the pileus center more flexuous, occasionally branched, thin-walled; terminal cells (12.9–)15.2–21.1–27(–32.3) × (2.2–)3.2–4.2–5.3(–6.6) μm, subcylindrical or clavate, apically usually obtuse, occasionally constricted; subterminal cells often shorter, rarely branched, ca. 2.5–6 μm wide. Pileocystidia near the pileus margin always long one-celled, (24.6–)37.3–55.6–73.8(–92.3) × (3.7–)4.8–5.9–7(–8.6) μm, cylindrical or clavate, apically usually obtuse or occasionally mucronate, thin-walled; contents dense granulose or crystalline, turning to reddish black in SV. Pileocystidia near the pileus center often longer, usually one-celled, occasionally three-celled, similar in shape and contents, (33.4–)50.3–70–89.6(–98) × (3.8–)4.6–5.3–6(–6.5) μm, thin-walled; contents turning to reddish black in SV.

Additional specimens examined: China, Yunnan province, Shangri-La City, Xiaozhongdian town, Shangjisha, atl. 3282 m, 27°27′32.01″ N, 99°48′52.44″ E, 22 August 2014, leg. Zhao2225 (RITF3123), GenBank number ITS: MW301627; ibid., 22 August 2014, leg. Zhao2211 (RITF3117), GenBank number ITS: MW301628.

Notes: Given the red pileus coloration, *Russula leucomarginata* could be confused with some taxa. It is mainly separated from similar-looking species by the internal transcribed spacers (ITS) sequence data. In our phylogenetic tree, *R**. leucomarginata* clustered with American species *R. rhodocephala* by 83% bootstrap support and formed a sister clade to *R. subsanguinaria* without support. However, *R. rhodocephala* is different from *R. leucomarginata* with a very firm basidiomata, a reddish orange pileus margin without striations, a strongly amyloid patch suprahilar spot, often longer basidia ((36.5–)41–45–49(–55) × (9–)10–11–12(–14) μm), and the association with *Pinus* [30]. *Russula subsanguinaria* differs from *R. leucomarginata* by its association with *Pinus*, a reddish orange to brownish red pileus margin, shorter hymenial cystidia on lamellae sides and marginal cells, and often longer pileocystidia. Similarly, European species *R. sanguinaria* is different from *R. leucomarginata*. *Russula sanguinaria* has a habitat mostly under *Pinus* and variable pileus diameter (27–120 mm), always smooth pileus margin, not peelable cuticle, fruity odor, often subacute to acute hymenial cystidia on lamellae sides, and usually shorter hymenial cystidia on lamellae edges (33–70 × 5–7.5 μm) [31,32].

***Russula roseola*** B. Chen, J. F. Liang and X. M. Jiang, sp. nov. (Figure 3C,D, Figure 4C,D, Figure 7 and Figure 8)

MycoBank: MB838149

Diagnosis: Mainly characterized by its reddish white to ruby red (12D8) in the pileus center, pink (11A4) to rose margin, adnate to slightly decurrent lamellae, often present lamellulae, reddish white to pink stipe, clavate or ellipsoid basidiola, and occasionally three-celled pileocystidia. The similarity of ITS sequence data of *R. roseola* is less than 97.5% with all similar-looking species.

Etymology: The Latin word “roseola” refers to the pink to rose pileus.

Type: China, Sichuan province, Meigu County, Dafengding National Nature Researve Hongxi Protection Station, 28°39′34.25″ N; 103°04′13.45″ E, 2560 m.a.s.l., in mixed coniferous forests, dominated by *Abies* and *Tsuga*, 11 July 2016, leg. Zhao Kuan DFD60 (RITF3418, holotype), GenBank number ITS: MW301618.

Description: Basidiomata small- to medium-sized. Pileus 30–60 mm in diam., planoconvex when young, broadly convex and with a depressed center when mature; margin slightly incurved, striation up to the 1/3 of the radius; surface smooth, glabrous, slightly viscous when humid, peeling readily; reddish white (11A2) to ruby red (12D8) in the pileus center; margin pink (11A4) to rose (12A5). Lamellae adnate to slightly decurrent, 2–3 mm in height, moderately distant, white (1A1); lamellulae often present and irregular in length, furcations absent or rare, edge entire and concolor. Stipe 50–70 × 5–8 mm, obclavate or cylindrical, expanded towards the base, rugulose longitudinally, reddish white (11A2) to pink (11A4), medulla hollow. Context 4–5 mm thick halfway to the margin in pileus, white, without color changing when bruised; taste mild to slightly acrid; odor indistinct. Spore print white to cream.

Basidiospores (5.6–)6.7–7.5–8.2(–8.9) × (5–)5.7–6.3–6.9(–7.7) μm, Q = (1.0–)1.08–1.18–1.29(–1.5), subglobose to broadly ellipsoid; ornamentation of large, moderately distant to dense (6–8(–10) in a 3 μm diam. circle) amyloid warts, 0.5–0.9 μm high, occasionally to frequently fused in short chains (1–2(–3) in the circle), rarely connected by line connections (0–1(–2) in the circle); suprahilar spot large, amyloid. Basidia (24.6–)36.8–43.2–49.6(–56.4) × (7.8–)11.5–13–14.5(–16.6) μm, mostly 4-spored, some 2- and 3-spored, broadly clavate; basidiola clavate or ellipsoid, ca. 7–14 μm wide. Hymenial cystidia on lamellae sides moderately numerous, ca. 1000–1400/mm^2^, (41.2–)43.4–58.2–73(–95) × (5.5–)6.3–8.5–10.6(–13.2) μm, clavate, fusiform or cylindrical, apically obtuse or acute, occasionally with 3–10 μm long appendage, thin-walled; contents abundant granulose or crystalline, turning to reddish black in SV. Hymenial cystidia on lamellae edges similar in size, (36.6–)41.8–54.6–67.4(–79.2) × (5.6–)7.4–9.9–12.3(–14.2) μm, clavate or cylindrical, apically obtuse or mucronate, sometimes with 2–6 μm long appendage, thin-walled; contents dense granulose or crystalline, turning to reddish black in SV. Marginal cells (10.2–)13.4–16.2–19(–22.3) × (2.8–)4–5.2–6.4(–8.4) μm, cylindrical or lageniform, sometimes flexuous. Pileipellis orthochromatic in Cresyl blue, sharply delimited from the underlying context, 100–140 μm deep, two-layered; suprapellis 50–70 μm deep, composed of ascending or erect, loose hyphal terminations; subpellis 60–80 μm deep, composed of horizontally oriented, relatively dense, intricate, 2–6 μm wide hyphae. Acid incrustations absent. Hyphal terminations near the pileus margin occasionally branched, often flexuous, thin-walled; terminal cells (9.4–)13.1–18–23.2(–30.3) × (2.4–)3.3–3.9–4.5(–5) μm, mainly cylindrical, occasionally clavate, apically obtuse; subterminal cells often wider, ca. 2.5–6 μm wide, usually unbranched. Hyphal terminations near the pileus center similar to those near the pileus margin; terminal cells (13.6–)15.6–20.5–25.4(–31) × 2.8–3.9–4.9(–7) μm, cylindrical or clavate, apically obtuse or occasionally constricted; subterminal cells often shorter, occasionally branched, ca. 3–5 μm wide. Pileocystidia near the pileus margin always long one-celled, occasionally three-celled, (43.6–)54.9–81.6–108(–133) × (4.2–)4.5–5.4–6.4(–8.3) μm, cylindrical or clavate, apically usually obtuse or occasionally mucronate, thin-walled; contents abundant granulose or crystalline, turning to tawny in SV. Pileocystidia near the pileus center often shorter, always long one-celled, occasionally two-celled, (31.2–)45.3–68.4–91(–111) × (3.8–)4.7–6.2–7.6(–9.5) μm, thin-walled; clavate or cylindrical, apically usually obtuse or occasionally mucronate, contents granulose, crystalline or flocculent, turning to reddish black in SV.

Additional specimens examined: China, Sichuan province, Meigu County, Dafengding National Nature Researve Hongxi Protection Station, 28°39′34.25″ N; 103°04′13.45″ E, atl. 2560 m, 11 July 2016, leg. Zhao Kuan DFD60-2(RITF3428), GenBank number ITS: MW301619; ibid., 11 July 2016, leg. Zhao Kuan DFD60-3(RITF3429), GenBank number ITS: MW301620.

Notes: *Russula roseola* is found in mixed coniferous forests dominated by *Abies* and *Tsuga* in China. It is separated from all similar-looking species by the internal transcribed spacers (ITS) sequence data. In the phylogenetic tree, the species is closely related to the North American species *R. americana*, Indian species *R. thindii*, American species *R. salishensis*, and European species *R. queletii* and *R. fuscorubroides*. However, *R. roseola* is distinct from them. *Russula americana* is also associated with *Abies* and *Tsuga*, but has larger basidiospores (8.5–11.5 × 7–10.8 μm) [31]. *Russula thindii* is distinct by its blood-red to fulvous pileus, reddish to vinaceous stipe, pale yellow spore print, larger basidiospores (7.5–9–10 × 6–7–8 μm), and hymenial cystidia on lamellae sides (55–123 × 12–15 μm) [33]. *Russula salishensis* resembles *R. roseola*, and it differs from our species by the association with both *Pseudotsuga menziesii* and *Tsuga heterophylla*, a deep wine red pileus center, a yellowish or pinkish to flesh-colored pileus margin with pale greenish grey, fruity odor, basidiospores with subreticulate ornamentation, and narrower basidia ((36.5–)41.5–46.5–52(–59.5) × (7.5–)8.5–9.5–10.5(–12) μm) [30]. *Russula queletii* mainly grows with *Picea* and possesses strong, fruity odor, and vinaceous red to violet pileus and stipe [13,34]. *Russula fuscorubroides* is usually found under *Picea* and can be distinguished by pileus of color binds dark wine to blackish crimson, reddish stipe, and larger basidiospores (8–10 × 6.5–8 μm) [34,35].

***Russula subsanguinaria*** B. Chen, J. F. Liang and X. M. Jiang, sp. nov. (Figure 3E,F, Figure 4E,F, Figure 9 and Figure 10)

MycoBank: MB838151

Diagnosis: Characterized by its reddish brown to dark brown pileus center, reddish orange to brownish red margin, reddish white to pink stipe, basidiospores with moderately distant to dense amyloid warts, reddish black hymenial cystidia in SV and sequence data.

Etymology: Subsanguinaria (Latin), named so because it is similar to *R.*
*sanguinaria*.

Type: China, Jilin province, An Tu County, Changbai Mountains, coniferous forest dominated by *Pinus*, 1000 m.a.s.l., 42°01′13″ N, 128°03′33″ E, 21 August 2012, leg. Zhang Xin 261 (RITF 2236, holotype), GenBank number ITS: MW301621.

*Description*: Basidiomata medium-sized. Pileus 55–70 mm in diam., first hemispheric when young, convex with a depressed center after maturation; margin incurved, rarely cracked, striation up to the 1/3 of the radius; surface smooth, glabrous, slightly viscous when humid, peeling readily; reddish brown (8D8) to dark brown (9F7) in the pileus center; margin reddish orange (7A7) to brownish red (8C8). Lamellae adnate to slightly subfree, 4–7 mm in height, moderately distant, white (1A1) to yellowish white (2A2); lamellulae sometimes present and irregular in length, furcations absent or rare, edge entire and concolor. Stipe 50–77 × 5–20 cm, central, obclavate to subcylindrical, expanded towards the base, surface dry, rugulose longitudinally, reddish white (7A2) to pink (8A3), medulla first solid, becoming hollow when mature. Context 2–3 mm thick halfway to the margin in pileus, white, unchanging when bruised; taste mild to slightly acrid; odor indistinct. Spore print cream to yellowish.

Basidiospores (6.3–)7.0–7.6–8.3(–9.3) × (5–)5.5–6.2–7(–7.7) μm, Q = (1.04)1.13–1.24–1.35(–1.47), subglobose to broadly ellipsoid to ellipsoid; ornamentation of large, moderately distant to dense (6–8(–10) in a 3 μm diam. circle) amyloid warts, 0.6–1(–1.2) μm high, mainly isolated, occasionally to frequently fused in short chains (1–3 fusions in the circle), rarely connected by line connections [0–1(–2) in the circle]; suprahilar spot large, amyloid. Basidia (25.4–)33–37.1–41.2(–42.6) × (5.6–)11–13.4–15.7(–17.7) μm, mostly 4-spored, some 2- and 3-spored, often broadly clavate, sometimes clavate; basidiola clavate or ellipsoid, ca. 7–15 μm wide. Hymenial cystidia on lamellae sides dispersed to moderately numerous, ca. 650–800/mm^2^, (39–)45.8–62.2–78.6(–122) × (5.3–)8.2–10.7–13.3(–16.5) μm, mainly broadly clavate, sometimes subcylindrical, apically often mucronate, rarely obtuse, always with 3–5 μm long appendage, thin-walled; contents abundant granulose or flocculent, turning to reddish black in SV. Hymenial cystidia on lamellae edges often smaller, (37.7–)41–55.7–70.5(–98) × (4.9–)6–8.7–11.4(–16.7) μm, clavate or subcylindrical, apically always acute or mucronate, rarely obtuse, usually with 5–8 μm long appendage, thin-walled; contents flocculent or granulose, turning to reddish black in SV. Marginal cells (10–)13.3–17.6–21.9(–28.3) × (2.3–)4–5.3–6.7(–8.9) μm, clavate or subcylindrical, sometimes flexuous. Pileipellis orthochromatic in Cresyl blue, sharply delimited from the underlying context, 120–180 μm deep, two-layered; suprapellis 80–110 μm deep, composed of ascending or erect, intricate hyphal terminations and pileocystidia; subpellis 60–80 μm deep, composed of horizontally oriented, dense, intricate, 3–7 μm wide hyphae. Hyphal terminations near the pileus margin scarcely branched, often flexuous, thin-walled; terminal cells (10–)15.8–21.5–27.2(–29.5) × (2.6–)3.1–4.1–5 (–6.2) μm, mainly subcylindrical, occasionally clavate, apically usually obtuse but occasionally constricted; subterminal cells usually shorter, ca. 3–7 μm wide, always unbranched. Hyphal terminations near the pileus center sometimes longer, not branched, usually flexuous, thin-walled; terminal cells (11.5–)18.1–26.1–34.1(–44.2) × (2.4–)3.3–4.3–5.3(–6.5) μm, lageniform or subcylindrical, apically obtuse or occasionally constricted; subterminal cells often wider, usually unbranched, ca. 3–6 μm wide. Pileocystidia near the pileus margin always long one-celled, (34.7–)43–60.9–78.7(–97.5) × (3.3–)3.9–5.1–6.3(–8.2) μm, cylindrical or narrowly clavate, apically always obtuse or scarcely mucronate, thin-walled; contents dense granulose, turning to brown in SV. Pileocystidia near the pileus center often longer (42.3–)60–82.1–104.2(–120) × (4.7–)5.3–6.2–7.2(–8.4) μm, cylindrical or narrowly clavate, apically usually obtuse, occasionally mucronate, thin--walled; contents dense granulose or crystalline, turning to brown in SV.

Additional specimens examined: China, Jilin province, An Tu County, Changbai Mountains, alt. 1000 m, 42°01′13″ N, 128°03′33″ E, 19 August 2012 leg. Zhang Xin 229 (RITF 2208), GenBank number ITS: MW301622; ibid., 19 August 2012, Zhang Xin 231 (RITF 2210), GenBank number ITS: MW301623; ibid., 21 July 2016, leg. CB6 (RITF 3442), GenBank number ITS: MW301624; ibid., 21 July 2016, leg. JXM126 (RITF 3435), GenBank number ITS: MW301625.

Notes: *Russula subsanguinaria* might be confused with *R. sanguinaria*, *R. americana*, and *R. rhodocephala*. It is separated from similar-looking species by the internal transcribed spacers (ITS) sequence data. Meanwhile, *Russula sanguinaria* differs in having bright carmine red pileus with smooth margin, fruity odor, and often larger hymenial cystidia on lamellae sides (63–100 × 6.3–14 μm) [31,32]. *Russula americana* is distinct by its frequently forked lamellae, larger basidiospore (8.5–11.5 × 7–10.8 μm), and association with *Abies* and *Tsu**ga* [31]. *Russula rhodocephala* has a very firm basidiomata, a bright scarlet pileus, a strongly amyloid patch suprahilar spot, and larger basidia ((36.5–)41–45–49(–55) × (9–)10–11–12(–14) μm) [30]. Additionally, *R. salishensis* resembles *R. subsanguinaria*, but it is distinct by the association with both *Pseudotsuga menziesii* and *Tsuga heterophylla*, a pileus margin with pale greenish grey, fruity odor, basidiospores with subreticulate ornamentation, and longer basidia ((36.5–)41.5–46.5–52(–59.5) × (7.5–)8.5–9.5–10.5(–12) μm) [30].

## Figures and Tables

**Figure 1 life-12-00480-f001:**
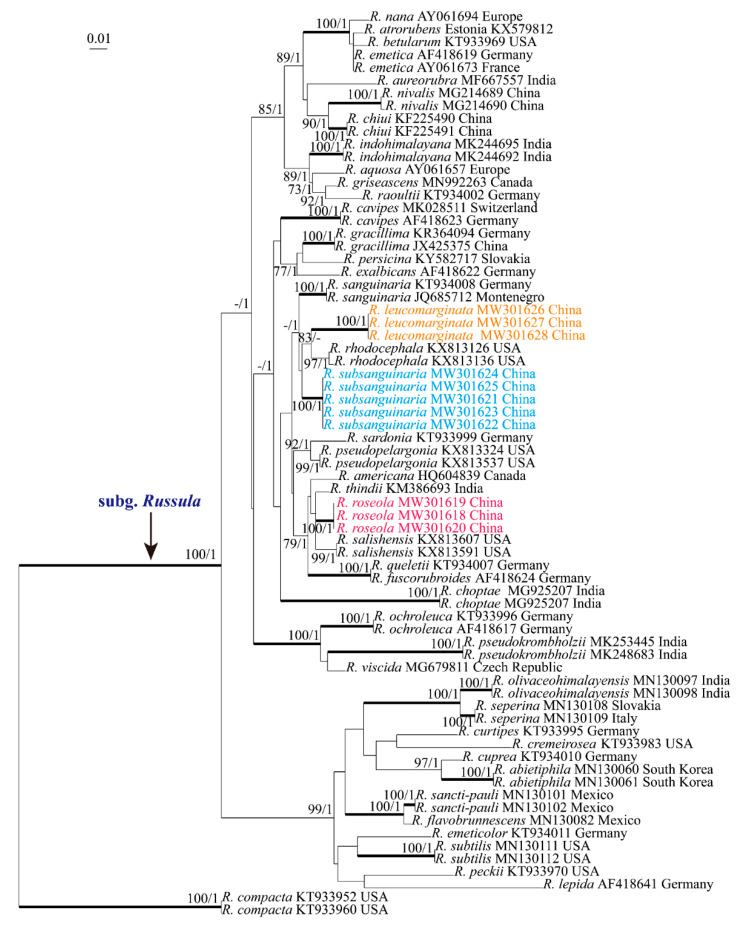
Phylogenetic tree based on the ITS sequence. Bootstrap Support (BS) ≥ 70% and Posterior Probabilities (PP) ≥ 0.95 are shown. Infrageneric classification of *Russula* follows Buyck et al. (2018).

**Figure 2 life-12-00480-f002:**
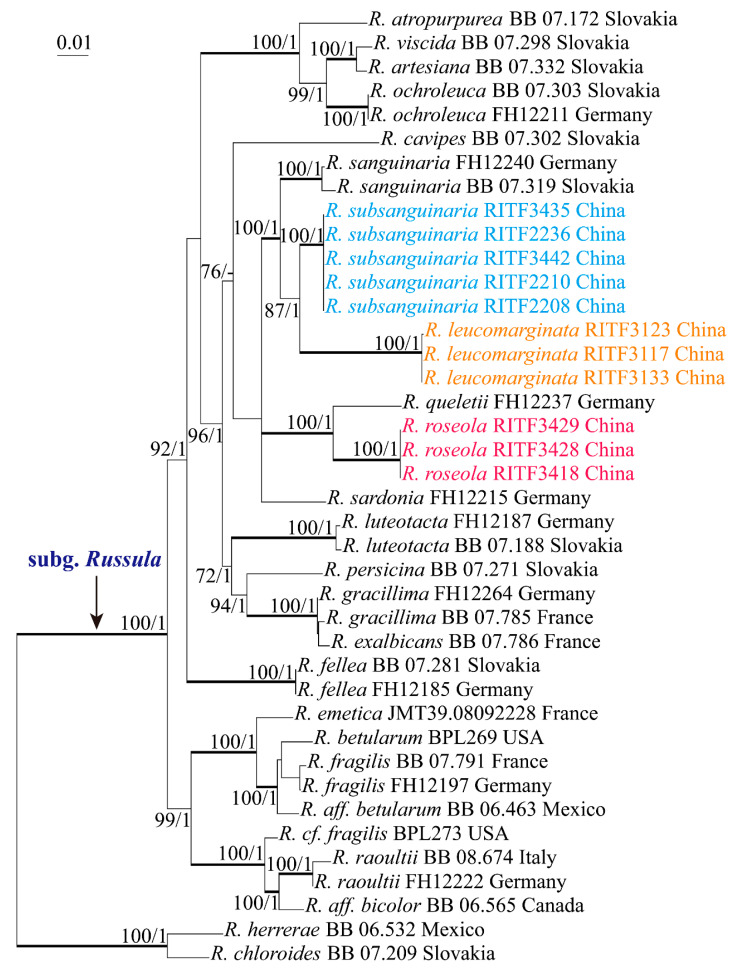
Phylogenetic tree based on the 28S-*RPB1*-*RPB2*-mtSSU dataset. Bootstrap support (BS, when ≥70%) and Bayesian posterior probabilities (PP, when ≥0.95) are shown. Infrageneric classification of *Russula* follows Buyck et al. (2018).

**Figure 3 life-12-00480-f003:**
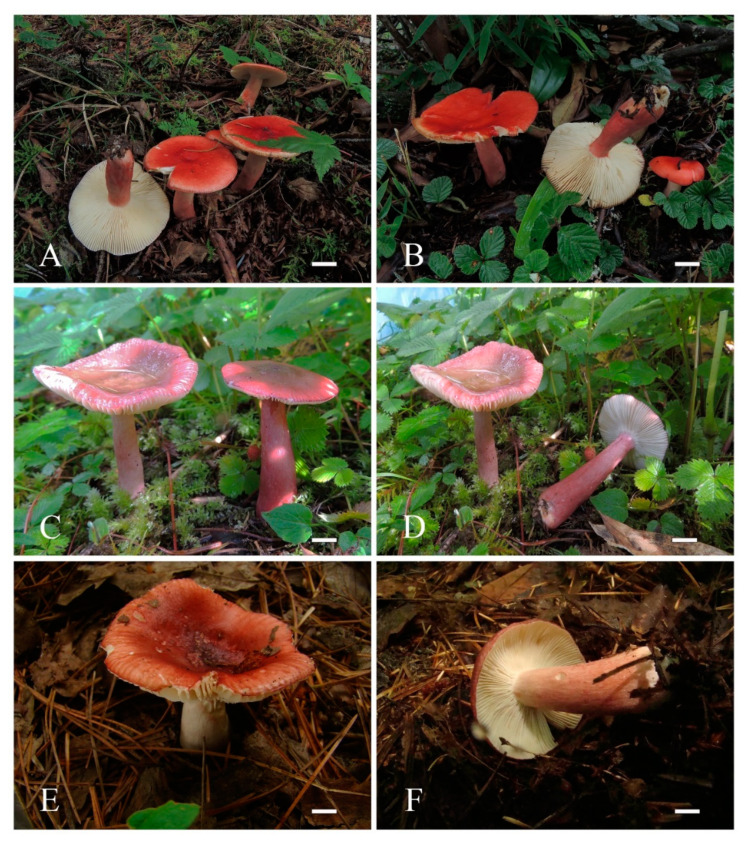
Basidiomata in the field. (**A**,**B**) *Russula leucomarginata* (RITF 3133, Holotype). (**C**,**D**) *Russula roseola* (RITF 3418, Holotype). (**E**,**F**) *Russula subsanguinaria* (RITF 2236, Holotype). Bars: 10 mm (**A**,**B**,**E**,**F**), 20 mm (**C**,**D**).

**Figure 4 life-12-00480-f004:**
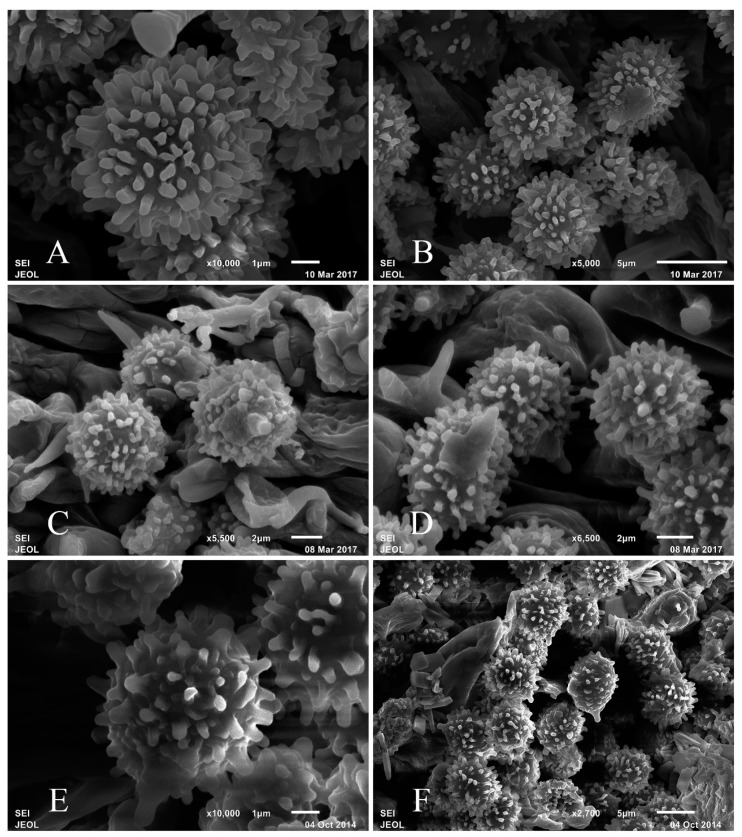
Basidiospores in SEM (JSM-6510LV). (**A**,**B**) *Russula leucomarginata* (RITF 3133, Holotype). (**C**,**D**) *Russula roseola* (RITF 3418, Holotype). (**E**,**F**) *Russula subsanguinaria* (RITF 2236, Holotype).

**Figure 5 life-12-00480-f005:**
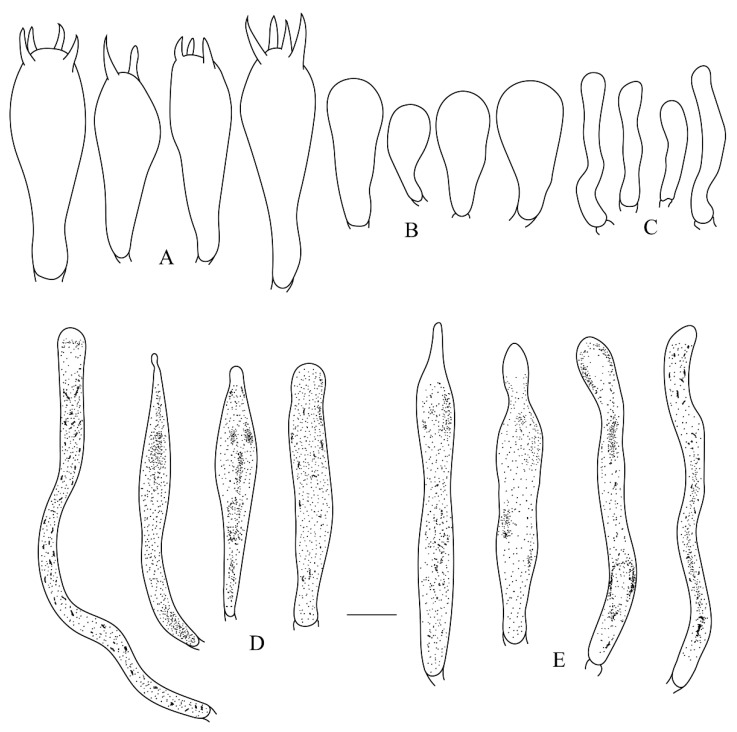
*Russula leucomarginata* (RITF 3133, Holotype). (**A**) Basidia. (B) Basidiola. (**C**) Marginal cells. (**D**) Hymenial cystidia on lamellae sides. (**E**) Hymenial cystidia on lamellae edges. Bar: 10 μm.

**Figure 6 life-12-00480-f006:**
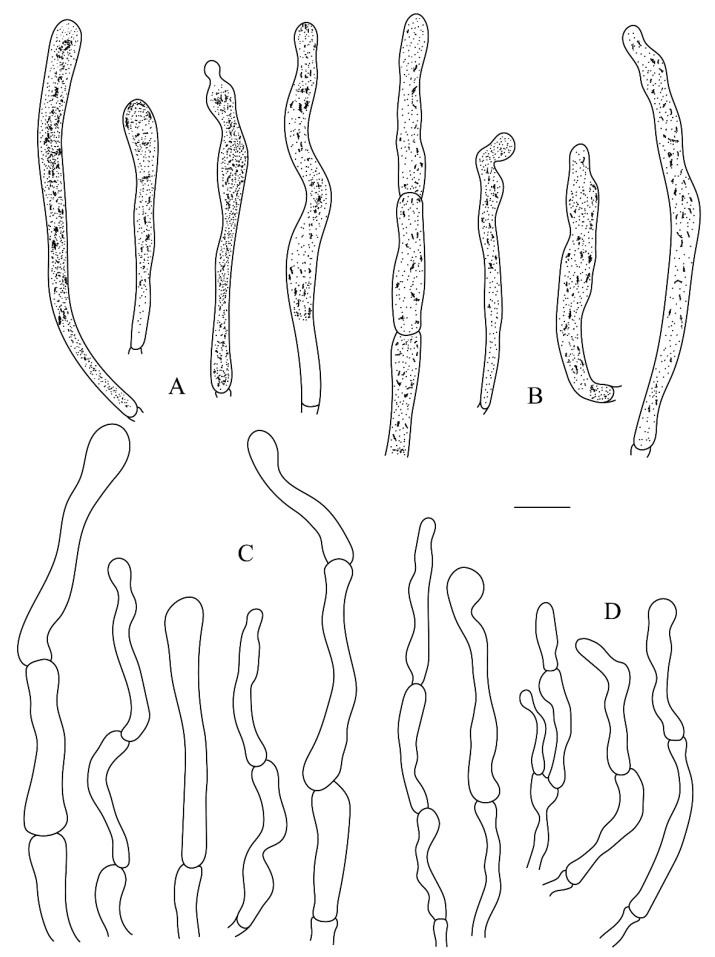
*Russula leucomarginata* (RITF 3133, Holotype). (**A**) Pileocystidia near the pileus margin. (**B**) Pileocystidia near the pileus center. (**C**) Hyphal terminations near the pileus margin. (**D**) Hyphal terminations near the pileus center. Bar: 10 μm.

**Figure 7 life-12-00480-f007:**
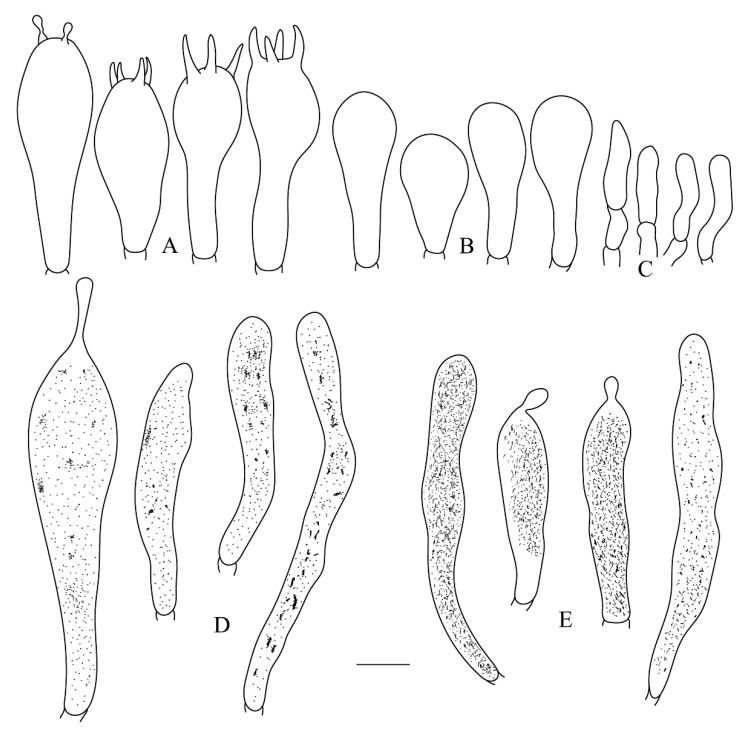
*Russula roseola* (RITF 3418, Holotype). (**A**) Basidia. (**B**) Basidiola. (**C**) Marginal cells. (**D**) Hymenial cystidia on lamellae sides. (**E**) Hymenial cystidia on lamellae edges. Bar: 10 μm.

**Figure 8 life-12-00480-f008:**
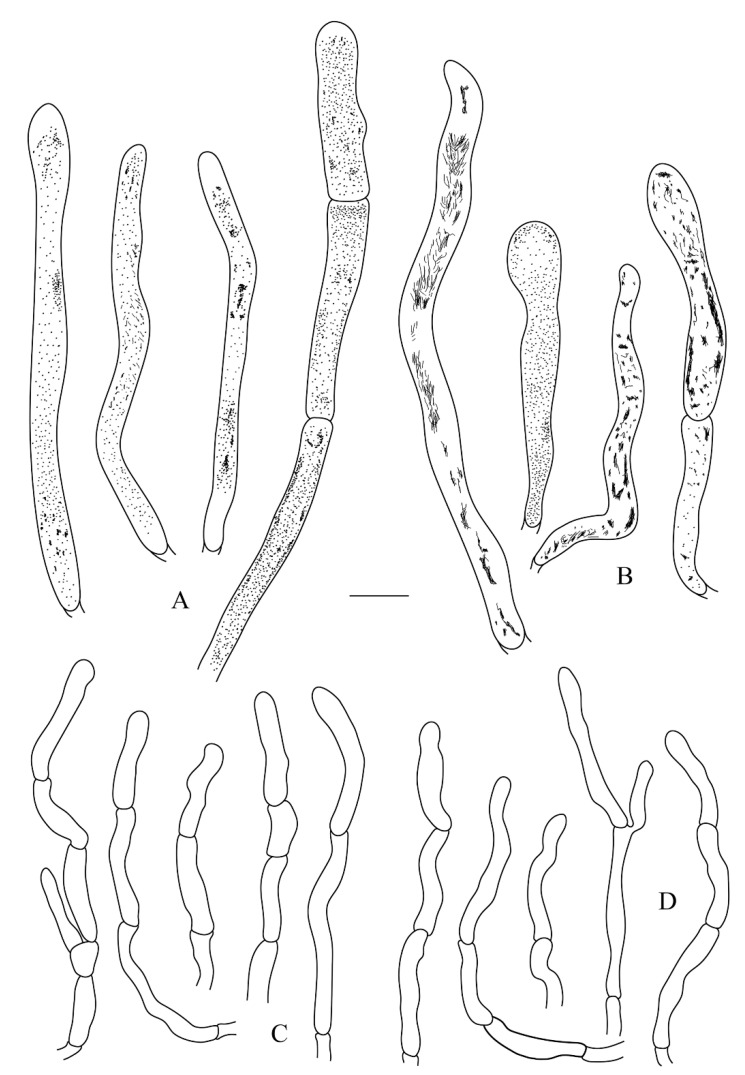
*Russula roseola* (RITF 3418, Holotype). (**A**) Pileocystidia near the pileus margin. (**B**) Pileocystidia near the pileus center. (**C**) Hyphal terminations near the pileus margin. (**D**) Hyphal terminations near the pileus center. Bar: 10 μm.

**Figure 9 life-12-00480-f009:**
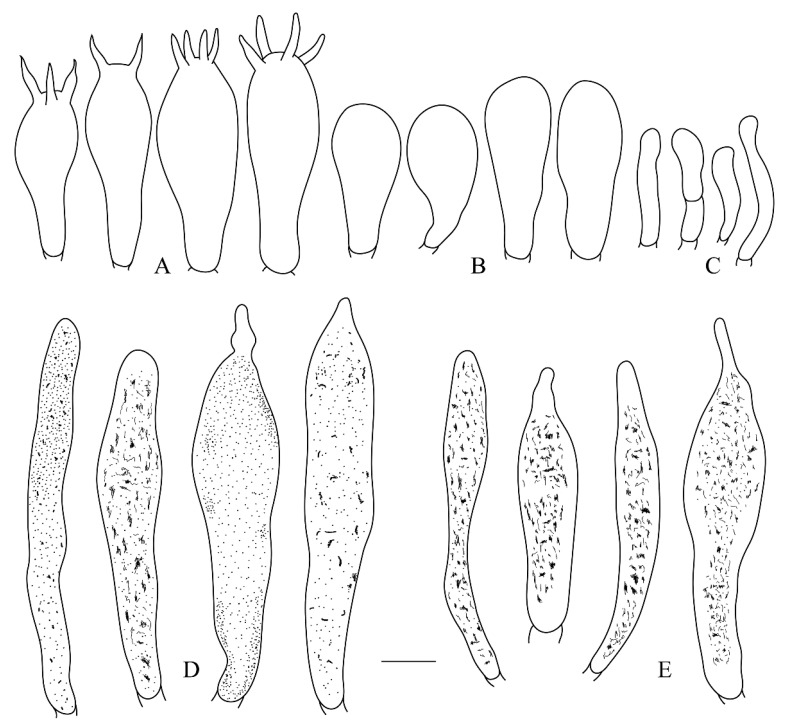
*Russula subsanguinaria* (RITF 2236, Holotype). (**A**) Basidia. (B) Basidiola. (**C**) Marginal cells. (**D**) Hymenial cystidia on lamellae sides. (**E**) Hymenial cystidia on lamellae edges. Bar: 10 μm.

**Figure 10 life-12-00480-f010:**
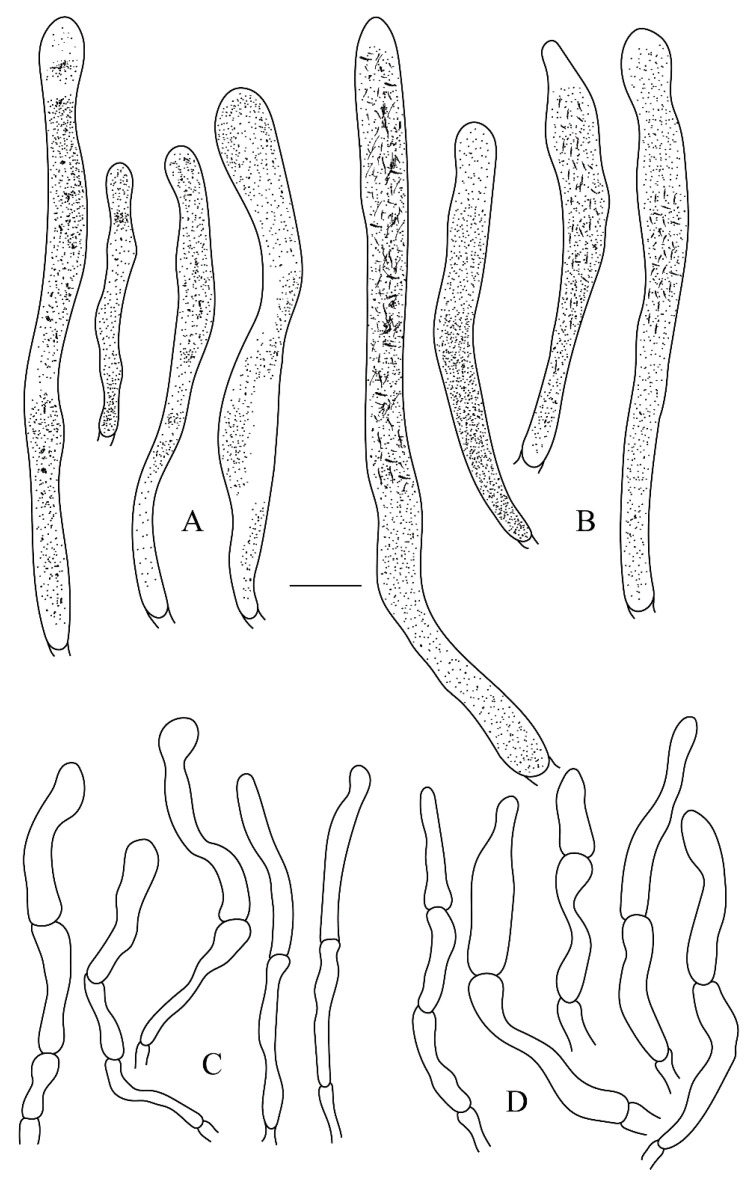
*Russula subsanguinaria* (RITF 2236, Holotype). (**A**) Pileocystidia near the pileus margin. (**B**) Pileocystidia near the pileus center. (**C**) Hyphal terminations near the pileus margin. (**D**) Hyphal terminations near the pileus center. Bar: 10 μm.

**Table 1 life-12-00480-t001:** GenBank accession numbers of sequences used in the multigene phylogenetic analysis. Newly generated sequences are in bold.

Taxon	Voucher	Location	28S	*RPB1*	*RPB2*	mtSSU	Reference
*R. aff. betularum*	BB 06.463	Mexico	KU237478	KU237627	KU237764	KU237322	[3]
*R. aff. bicolor*	BB 06.565	Canada	KU237477	KU237626	KU237763	KU237321	[3]
*R. artesiana*	BB 07.332	Slovakia	KU237511	KU237357	KU237797	KU237357	[3]
*R. atropurpurea*	BB 07.172	Slovakia	KU237550	KU237696	KU237836	KU237396	[3]
*R. betularum*	BPL269	USA	KT933829	KT957341	KT933900	‒	[12]
*R. cavipes*	BB 07.302	Slovakia	KU237554	KU237698	KU237840	KU237400	[3]
*R.* cf. *fragilis*	BPL273	USA	KT933833	KT957345	KT933904	‒	[12]
*R. chloroides*	BB 07.209	Slovakia	KU237559	KU237703	KU237845	KU237407	[3]
*R. emetica*	JMT39.08092228	France	KU237578	KU237721	KU237864	KU237426	[3]
*R. exalbicans*	BB 07.786	France	KU237568	KU237712	KU237854	KU237416	[3]
*R. fellea*	BB 07.281	Slovakia	KU237507	KU237656	KU237793	KU237352	[3]
*R. fellea*	FH12185	Germany	KT933850	KT957361	KT933921	‒	[12]
*R. fragilis*	FH12197	Germany	KT933854	KT957365	KT933925	‒	[12]
*R. fragilis*	BB 07.791	France	KU237506	KU237655	KU237792	KU237351	[3]
*R. gracillima*	FH12264	Germany	KR364226	KR364472	KR364342	‒	[28]
*R. gracillima*	BB 07.785	France	KU237504	KU237653	KU237790	KU237349	[3]
*R. herrerae*	BB 06.532	Mexico	KU237486	KU237635	KU237772	KU237330	[3]
** *R. leucomarginata* **	**RITF3133**	**China**	**MW309327**	**MW310557**	**MW310568**	**MW309338**	**This work**
** *R. leucomarginata* **	**RITF3123**	**China**	**MW309328**	**MW310558**	**MW310569**	**MW309339**	**This work**
** *R. leucomarginata* **	**RITF3117**	**China**	**MW309329**	**MW310559**	**MW310570**	**MW309340**	**This work**
*R. luteotacta*	BB 07.188	Slovakia	KU237512	KU237662	KU237798	KU237358	[3]
*R. luteotacta*	FH12187	Germany	KT933852	KT957363	KT933923	‒	[12]
*R. ochroleuca*	FH12211	Germany	KT933857	KT957368	KT933928	‒	[12]
*R. ochroleuca*	BB 07.303	Slovakia	KU237519	KU237669	KU237805	KU237365	[3]
*R. persicina*	BB 07.271	Slovakia	KU237494	KU237642	KU237780	KU237338	[3]
*R. queletii*	FH12237	Germany	KT933868	KT957378	KT933939	‒	[12]
*R. raoultii*	BB 08.674	Italy	KU237551	KU237697	KU237837	KU237397	[3]
*R. raoultii*	FH12222	Germany	KT933863	KT957373	KT933934	‒	[3]
** *R. roseola* **	**RITF3418**	**China**	**MW309319**	**‒ **	**MW310560**	**MW309330**	**This work**
** *R. roseola* **	**RITF3428**	**China**	**MW309320**	**‒ **	**MW310561**	**MW309331**	**This work**
** *R. roseola* **	**RITF3429**	**China**	**MW309321**	**‒ **	**MW310562**	**MW309332**	**This work**
*R. sanguinaria*	BB 07.319	Slovakia	KU237503	KU237652	KU237789	KU237348	[3]
*R. sanguinaria*	FH12240	Germany	KT933869	KT957379	KT933940	‒	[12]
*R. sardonia*	FH12215	Germany	KT933860	KT957371	KT933931	‒	[12]
** *R. subsanguinaria* **	**RITF2236**	**China**	**MW309322**	**MW310552**	**MW310563**	**MW309333**	**This work**
** *R. subsanguinaria* **	**RITF2208**	**China**	**MW309323**	**MW310553**	**MW310564**	**MW309334**	**This work**
** *R. subsanguinaria* **	**RITF2210**	**China**	**MW309324**	**MW310554**	**MW310565**	**MW309335**	**This work**
** *R. subsanguinaria* **	**RITF3442**	**China**	**MW309325**	**MW310555**	**MW310566**	**MW309336**	**This work**
** *R. subsanguinaria* **	**RITF3435**	**China**	**MW309326**	**MW310556**	**MW310567**	**MW309337**	**This work**
*R. viscida*	BB 07.298	Slovakia	KU237491	KU237639	KU237777	KU237335	[3]

## Data Availability

All data obtained and analyzed in this study have been included in this article.

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
