# Peer review of "Morphological Characters and Molecular Phylogeny Reveal Three New Species of Subgenus Russula from China"

_life, 2022, doi:10.3390/life12040480_

Round 1

Reviewer 1 Report

There are still some language issues that need to be resolved. These are indicated on the annotated manuscript.

Author Response

  1. This sentence should be looked at. Perhaps it was to be merged with the previous sentence

Response: Done. We have revised it.

Given the red pileus coloration, Russula leucomarginata could be confused with some taxa. It is mainly separated from similar looking species by the internal transcribed spacers (ITS) sequence data.

  1. Again this is an not a good contruction

Response: Done. We have revised it.

Russula rhodocephala is different from R. leucomarginata with a very firm basidiomata, a reddish orange pileus margin without striations, a strongly amyloid patch suprahilar spot, often longer basidia [(36.5–)41–45–49(–55) × (9–)10–11–12(–14) μm] and the association with Pinus

  1. something missing here

Response: Done.

we measured the thickness of context at halfway the pileus radius.

  1. merge with the previous sentence

Response: Done. We have revised it.

It is separated from all similarly-looking species by the internal transcribed spacers (ITS) sequence data.

  1. Sentence should start with the word spelled-out. Remove with in line two.

Response: Done. We have revised it.

Russula salishensis resembles R. roseola, it differs from our species by the association with both Pseudotsuga menziesii and Tsuga heterophylla,

  1. Clarify

Response: Done. We have revised it.

It is separated from similar looking species by the internal transcribed spacers (ITS) sequence data.

  1. remove "with"

Response: Done. We have removed "with".

Additionally, R. salishensis resembles R. subsanguinaria,……

Reviewer 2 Report

The authors have already been informed that sanguinea and sanguinaria are the same species... they should check which one (preferably sanguinaria) they are going to use and change perhaps 'subsanguinea sp nov' into 'subsanguinaria' sp nov

other remarks in attached

Author Response

  1. The authors have already been informed that sanguinea and sanguinaria are the same species... they should check which one (preferably sanguinaria) they are going to use and change perhaps 'subsanguinea sp nov' into 'subsanguinaria' sp nov

Response: Done. We have contacted MycoBank curators to modify the name. In the text, we have replaced the epithet " subsanguinea " with " subsanguinaria ".

  1. delete.... this is not a character for this species but for most basidio

Response: Done. We have deleteg it.

Russula leucomarginata is recognized by a yellowish red to reddish brown pileus center, a yellowish white to reddish white and sometimes cracked margin and a reddish white to pastel pink stipe.

  1. Many systems proposing an infrageneric classification of the genus Russula have been proposed in the past few years

Response: Done. We have revised it.

  1. I already told you that these are synonyms... it is the same species use sanguinaria

Response: Done. In the text, we have replaced the epithet " subsanguinea " with " subsanguinaria ".

  1. you can put obtained sequence data

Response: Done. We have put obtained sequence data in diagnosis.

This manuscript is a resubmission of an earlier submission. The following is a list of the peer review reports and author responses from that submission.

Round 1

Reviewer 1 Report

The manuscript needs to be thorough reviewed by and English speaker. There are far too many corrects for a reviewer to make.

There is the use of the work "oder" in several places particularly in the abstract. I do not know what is meant.

The abstract is poorly written and includes words such as: "basdiola" presumably meaning basidiome, "oder", "photo brown"

The descriptions are well done and seem to have been written by someone other than the writer of the abstract and introduction.

The importance of the paper is that it further documents the diversity in the genus Russula.

Author Response

The following are the answers and revisions I have made in response to reviewer' questions and suggestions. The modified section is set against a blue background in the text.

1.The manuscript needs to be thorough reviewed by and English speaker. There are far too many corrects for a reviewer to make.

Response: Done. We have checked and revised the manuscript throughout.

2.There is the use of the work "oder" in several places particularly in the abstract. I do not know what is meant.

Response: Done. We have replaced “oder pinkish” with “reddish white”.

3.The abstract is poorly written and includes words such as: "basdiola" presumably meaning basidiome, "oder", "photo brown"

Response: Done. We have replaced “the words” with “new words”.

4.The descriptions are well done and seem to have been written by someone other than the writer of the abstract and introduction.

Response: Done. The original manuscript is written by the first author.

5.The importance of the paper is that it further documents the diversity in the genus Russula.

Response: Done. In order to better understand the biological diversity and geographic distribution of Chinese subgenus Russula, extensive survey have been undertaken in different parts of China. Here, three taxa are proposed as new to science.

Reviewer 2 Report

Many of typographic or grammatical mistakes are just highlighted in yellow the pdf, but more are probably present and language should be checked (esp the presence of 'oder' is unexplainable)

the main problem of the paper is no longer the inclusion of close species (as it was in the first version), but the comparison with these close species. Also the diagnoses are almost irrelevant (none of the diagnoses allows to recognize a species)... many species in this core clade are rather host-specific and sequence data are the most important in the diagnosis (I would like to see % similarity with similarly-looking species in the diagnoses). Also, comments on individual new taxa should refer and discuss comments to species in Sarnari, as well as in Bazzicalluppo et al. 2017 (Fungal diversity), now these comments are mostly restricted to citation of minor differences in measurements... which are in nearly all cases based upon a single fruiting body that was studied for the description.

Author Response

The following are the answers and revisions I have made in response to reviewer' questions and suggestions. The modified section is set against a blue background in the text.

1.Many of typographic or grammatical mistakes are just highlighted in yellow the pdf, but more are probably present and language should be checked (esp the presence of 'oder' is unexplainable)

Response: Done. We have checked and revised the manuscript throughout. And We have replaced “the words” with “new words”.

  1. the main problem of the paper is no longer the inclusion of close species (as it was in the first version), but the comparison with these close species. Also the diagnoses are almost irrelevant (none of the diagnoses allows to recognize a species)... many species in this core clade are rather host-specific and sequence data are the most important in the diagnosis (I would like to see % similarity with similarly-looking species in the diagnoses). Also, comments on individual new taxa should refer and discuss comments to species in Sarnari, as well as in Bazzicalluppo et al. 2017 (Fungal diversity), now these comments are mostly restricted to citation of minor differences in measurements... which are in nearly all cases based upon a single fruiting body that was studied for the description.

Response: Done. We have added the similarity of ITS sequence data of our species with similarly-looking species in the diagnoses. And We have added some new discuss in the “note”

Russula leucomargola

Diagnosis: Recognized by its cracked pileus, yellowish red to reddish brown pileus center, yellowish white to reddish margin, reddish white to pastel pink stipe, clavate or ellipsoid basidiola. It is mainly separated from similar looking species by the internal transcribed spacers (ITS) sequence data. And the similarity is less than 97%with these species.

Russula roseola

Diagnosis: Mainly characterized by its reddish white to ruby red (12D8) in the pileus center, pink (11A4) to rose margin, adnate to slightly decurrent lamellae, often present lamellulae, reddish white to pink stipe, clavate or ellipsoid basidiola and occasionally three-celled pileocystidia. And the similarity of ITS sequence data of R. roseola is less than 97.5% with all similarly-looking species.

Russula subsanguinea

Diagnosis: Characterized by its reddish brown to dark brown pileus center, reddish orange to brownish red margin, reddish white to pink stipe, basidiospores with moderately distant to dense amyloid warts, reddish black hymenial cystidia in SV. And it is mainly separated from similar looking species by the internal transcribed spacers (ITS) sequence data.

  1. not really a hot spot yet... this is just because so few Russula have been described before now... it is just 'catching up' for the moment.

Response: Done. We have revised the sentence in the manuscript.

Recently, some new species of subgenus Russula have been successively reported in Asia, indicating that many unknown Russula species are waiting to explore.

  1. all the discussed species in the following paragraph are red! So quite similar host might be more important (what are hosts of your other collections?)

Response: Done. The hosts of other collections are same as holotype.

  1. from Pacific North West, where many sister taxa occurr for Asian species. How similar is ITS with your species?

Response: Done. The similarity of ITS sequence of our species with Pacific North West species is less than 96%.

  1. again, these are all brick red species... host and sequence data might help to distinguish , but R. sanguinea is also Pinus associated, and so is R. rhodocephala (but not R. americana)

Response: Done. We have added the similarity of ITS sequence data of our species with similarly-looking species in the diagnoses.

Reviewer 3 Report

This manuscript has the potential to become an interesting contribution to the taxonomy of Russula from China. However, it suffers from major flaws/problems. First, the phylogenetic analyses are quick and dirty and poorly described (both in methods and results). The following should be done:

  1. Perform partitioned analyses (both for ML and Bayesian).
  2. I do not see why ITS is not included in the multigene phylogeny
  3. More details should be given in the methods, especially for Bayesian inference. Each phylogenetic analysis is different, depending on the dataset.
  4. The proportion of sites retained after Gblock analysis should be given
  5. The alignments should be made available to reviewers (the TreeBase accession number is not enough, because the data are not available prior to publication of the paper). Alignments prior to running Gblocks should be given, because the quality of the alignment cannot be properly assessed after sites have been eliminated by Gblock.
  6. The statement in line 124-125 is not correct. The analyses did not show that Russula subgen. was monophyletic. It was monophyletic because it was the ingroup opposed to an outgroup composed of only 2 species. By the way, how was this outgroup selected?

Morphological descriptions and illustrations should be improved.

  1. Illustrations of terminations of hyphae from the pileipellis centre and margin are missing
  2. Spores are poorly illustrated. The SEM micrographs of R. roseola spores are particularly poor. Moreover, how do the spores look like using compound light microscope ? Not everybody has access to SEM...
  3. Color changes with SV should be illustrated too
  4. All the diagnoses refer to the color of the pileipellis. Based on my experience of Russula, pileus colors can be quite variable, so I think diagnoses should not refer to colors, especially if only different shades of red. As a reminder, diagnoses should not be a summary of the description, but should point out the characters that allow differentiating the new species from the most closely resembling species.

The English is quite poor, and even more problematic is the misuse of scientific terms. For example, the use of "crown clade", as opposed to "core clade" is not correct. Another example is the use of the obsolete term "fruiting bodies". Basidiomes (or basidiomata) should be used instead. Please be careful about the terminology.

The name leucomargola sounds really weird to me. I think leucomarginata would be better.

Author Response

The following are the answers and revisions I have made in response to reviewer' questions and suggestions. The modified section is set against a blue background in the text.

This manuscript has the potential to become an interesting contribution to the taxonomy of Russula from China. However, it suffers from major flaws/problems. First, the phylogenetic analyses are quick and dirty and poorly described (both in methods and results). The following should be done:

  1. Perform partitioned analyses (both for ML and Bayesian).

Response: Done. we performed phylogenetic analysis based on ITS sequence and 28S-RPB1-RPB2-mtSSU datasets. And the phylogenetic analyses results are consistent.

  1. I do not see why ITS is not included in the multigene phylogeny
  2. Response: Done. Some species only have ITS sequence data which are closely related to our new species. If ITS is included in the multigene phylogeny, there are many empty loci. The phylogenetic tree could be inaccurate. So we respectively performed phylogenetic analysis based on ITS sequence and 28S-RPB1-RPB2-mtSSU datasets. And we ensure that ITS phylogeny and multigene phylogeny are consistent.
  3. More details should be given in the methods, especially for Bayesian inference. Each phylogenetic analysis is different, depending on the dataset.

Response: Done. We have added more details on ML analysis.

The analysis was executed by applying the rapid bootstrap algorithm with 1000 replicates to affirm the consistency of the results under the GAMMA model. Only the maximum-likelihood best tree from all searches was kept.

For BA analyses, the GTR model was selected as the best substitution model by MrModeltest. A rapid bootstrapping was undertaken to affirm the consistency of results with 1000 replicates under the GTRGAMMA model.BA was done with MrBayes on XSEDE (3.2.7a) through the Cipres Science Gateway (www.phylo.org) under the GTR model. Four Markov chains were run for a total of 50,000,000 generations and trees were sampled every 100 generations, the first 25% of each sampled topology being discarded as part of the burn-in procedure.

  1. The proportion of sites retained after Gblock analysis should be given

Response: Done. To obtain reliable and reasonable results, the online program Gblocks 0.91b (Gblocks_server.html) was used in default parameters, approximately 77.8% sites retained after Gblock analysis

  1. The alignments should be made available to reviewers (the TreeBase accession number is not enough, because the data are not available prior to publication of the paper). Alignments prior to running Gblocks should be given, because the quality of the alignment cannot be properly assessed after sites have been eliminated by Gblock.

Response: Done. We have submitied the final sequence alignment to TreeBASE (http://purl.org/phylo/treebase/phylows/study/TB2: S28485).

  1. The statement in line 124-125 is not correct. The analyses did not show that Russula subgen. was monophyletic. It was monophyletic because it was the ingroup opposed to an outgroup composed of only 2 species. By the way, how was this outgroup selected?

Response: Done. We have revised it.

The ITS phylogenetic analysis showed that subg. Russula obtain a high support (BS 100%, PP 1)

We selected the two outgroups basing the genus Russula phylogenetic analysis.  

Morphological descriptions and illustrations should be improved.

  1. Illustrations of terminations of hyphae from the pileipellis centre and margin are missing

Response: Done. Micromorphological features were described and illustrated referring to Adamčík et al 2019 (The quest for a globally comprehensible Russula language. Fungal Divers 2019, 99, 369–449).

  1. Spores are poorly illustrated. The SEM micrographs of R. roseola spores are particularly poor. Moreover, how do the spores look like using compound light microscope ? Not everybody has access to SEM...

Response: Done. Size, shape and color changes and of Spores are have been accurately described in the text using compound light microscope.

  1. Color changes with SV should be illustrated too

Response: Done. Color changes with SV have been accurately described in the text.

  1. All the diagnoses refer to the color of the pileipellis. Based on my experience of Russula, pileus colors can be quite variable, so I think diagnoses should not refer to colors, especially if only different shades of red. As a reminder, diagnoses should not be a summary of the description, but should point out the characters that allow differentiating the new species from the most closely resembling species.

Response: Done. We have added the similarity of ITS sequence data of our species with similarly-looking species in the diagnoses.

Russula leucomargola

Diagnosis: Recognized by its cracked pileus, yellowish red to reddish brown pileus center, yellowish white to reddish margin, reddish white to pastel pink stipe, clavate or ellipsoid basidiola. It is mainly separated from similar looking species by the internal transcribed spacers (ITS) sequence data. And the similarity is less than 97%with these species.

Russula roseola

Diagnosis: Mainly characterized by its reddish white to ruby red (12D8) in the pileus center, pink (11A4) to rose margin, adnate to slightly decurrent lamellae, often present lamellulae, reddish white to pink stipe, clavate or ellipsoid basidiola and occasionally three-celled pileocystidia. And the similarity of ITS sequence data of R. roseola is less than 97.5% with all similarly-looking species.

Russula subsanguinea

Diagnosis: Characterized by its reddish brown to dark brown pileus center, reddish orange to brownish red margin, reddish white to pink stipe, basidiospores with moderately distant to dense amyloid warts, reddish black hymenial cystidia in SV. And it is mainly separated from similar looking species by the internal transcribed spacers (ITS) sequence data.

The English is quite poor, and even more problematic is the misuse of scientific terms. For example, the use of "crown clade", as opposed to "core clade" is not correct. Another example is the use of the obsolete term "fruiting bodies". Basidiomes (or basidiomata) should be used instead. Please be careful about the terminology.

Response: Done. Response: Done. We have checked and revised the manuscript throughout. And we have replaced “the words” with “new words”. And the "crown clade" and "core clade" are in accordance with Adamčík et al.2019. (The quest for a globally comprehensible Russula language)

The name leucomargola sounds really weird to me. I think leucomarginata would be better.

Response: Done. leuco (Latin) = white; margola (Latin) = margin; named after the yellowish white to reddish white margin of pileus. And We have submitted the name to MycoBank.

Round 2

Reviewer 1 Report

Only a couple of minor things noted on the manuscript.

Reviewer 3 Report

Most of my comments were not properly addressed. The authors should have more respect for reviewers' work on their manuscript and make the effort to properly address their comments. They can refute comments, but only with good and correct arguments, which is not the case here. Phylogenetic analyses are still not described/performed well enough, and using awkward wording. The argument not to include ITS in the multigene analyses just does not make any sense. You simply do not have to include those accessions for which only ITS is available... The term "crown clade" is not correctly used. It is not because someone else published a paper with the same use of the term that it is correct. Mistakes are published all the time, even in Fungal Diversity. A crown clade refers to a clade that includes all extant (living) organisms (terminal nodes or leaves) as well as all the nodes down to the most recent common ancestor of the terminal nodes. It is opposed to a stem clade, which contains taxa that appeared during the evolution towards the crown clade, but went extinct. So the "core clade" in the ms. is also a crown clade.

Adding the ITS similarity in diagnoses reflects the fact that the authors still do not understand what is a diagnosis... It would be like saying about morphology, for example "pileus color different, and spores of a different shape". If you want to include molecular character in a diagnosis, you to specify what precisely are the differences between the species, e.g. AcctgtaG instaead of GcctgtaA at positions 211-218. To do that, however, you have to be quite sure of the diagnostic value of those characters, i.e., that the differences can be observed in all individuals of the species. That would require quite extensive sampling...

The epithet "leucomargola" is wrong and should not be used. Check your botanical latin, or have it checked by someone who knows. You can contact MycoBank curators to modify the name or delete the incorrect name.